

# Acoustic monitoring indicates a correlation between calling and spawning in captive spotted seatrout (*Cynoscion nebulosus*)

Eric W. Montie[1], Matt Hoover[2], Christopher Kehrer[1], Justin Yost[3], Karl Brenkert[3], Tim O'Donnell[3] and Michael R. Denson[3]

[1] Department of Natural Sciences, University of South Carolina Beaufort, Bluffton, SC, United States
[2] Department of Computational Science and Mathematics, University of South Carolina Beaufort, Bluffton, SC, United States
[3] South Carolina Department of Natural Resources, Marine Resources Research Institute, Charleston, SC, United States

Corresponding author
Eric W. Montie, emontie@uscb.edu

## ABSTRACT

**Background**. Fish sound production is widespread throughout many families. Territorial displays and courtship are the most common reasons for fish sound production. Yet, there is still some questions on how acoustic signaling and reproduction are correlated in many sound-producing species. In the present study, our aim was to determine if a quantitative relationship exists between calling and egg deposition in captive spotted seatrout (*Cynoscion nebulosus*). This type of data is essential if passive acoustics is to be used to identify spawning aggregations over large spatial scales and monitor reproductive activity over annual and decadal timeframes.

**Methods**. Acoustic recorders (i.e., DSG-Oceans) were placed in three laboratory tanks to record underwater sound over an entire, simulated reproductive season. We enumerated the number of calls, calculated the received sound pressure level, and counted the number of eggs every morning in each tank.

**Results**. Spotted seatrout produced three distinct call types characterized as "drums," "grunts," and "staccatos." Spotted seatrout calling increased as the light cycle shifted from 13.5 to 14.5 h of light, and the temperature increased to 27.7 °C. Calling decreased once the temperature fell below 27.7 °C, and the light cycle shifted to 12 h of light. These temperature and light patterns followed the natural reproductive season observed in wild spotted seatrout in the Southeast United States. Spotted seatrout exhibited daily rhythms in calling. Acoustic signaling began once the lights turned off, and calling reached maximum activity approximately 3 h later. Eggs were released only on evenings in which spotted seatrout were calling. In all tanks, spotted seatrout were more likely to spawn when male fish called more frequently. A positive relationship between SPL and the number of eggs collected was found in Tanks 1 and 3.

**Discussion**. Our findings indicate that acoustic metrics can predict spawning potential. These findings are important because plankton tows may not accurately reflect spawning locations since egg capture is likely affected by predator activity and water currents. Instead, passive acoustics could be used to monitor spotted seatrout reproduction. Future studies can use this captive study as a model to record the estuarine soundscape precisely over long time periods to better understand how human-made stressors (e.g., climate change, noise pollution, and chemical pollutants) may affect spawning patterns.

## INTRODUCTION

Many fish families contain species that produce sounds for communication purposes. Some examples include African freshwater fishes (Mormyridae), damselfishes (Pomacentridae), grouper (Serranidae), catfishes (Ictaluridae, Pimelodidae, Doradidae, and Mochokidae), rockfishes (Sebastidae), haddock and cod (Gadidae), triglids (Triglidae), cichlids (Cichlidae), toadfishes (Batrachoididae), and drums (Sciaenidae) (e.g., *Tavolga, 1958*; *Holt, Holt & Arnold, 1985*; *Fine, Burns & Harris, 1990*; *Mann & Lobel, 1995*; *Amorim, 2006*; *Rowe & Hutchings, 2006*; *Luczkovich et al., 2008*; *Mann & Grothues, 2009*; *Parmentier et al., 2009*; *Walters et al., 2009*; *Širović et al., 2009*; *Mann et al., 2010*; *Locascio & Mann, 2011*). One of the primary functions of sound production in fishes is for courtship during the reproductive season.

The Family Sciaenidae contains fish renowned for producing sound. Some representative examples include the Atlantic croaker (*Micropogonias undulatus*), red drum (*Sciaenops ocellatus*), weakfish (*Cynoscion regalis*), American star drum (*Stellifer lanceolatus*), black drum (*Pogonias cromis*), silver perch (*Bairdiella chrysoura*), and spotted seatrout (*Cynoscion nebulosus*) (e.g., *Hill, Michael & Musick, 1987*; *Nieland & Wilson, 1993*; *Sprague et al., 2000*; *Collins, Callahan & Post, 2001*; *Montie, Vega & Powell, 2015*). Sciaenids most likely evolved the ability to produce acoustic signals in order to communicate in turbid estuaries where the water visibility is minimal (*Holt, Godbout & Arnold, 1981*; *Holt, 2008*). In this family, fish produce sound by contracting a sonic muscle against an inflated swim bladder (reviewed in *Fine & Parmentier, 2015*). In most Sciaenids, males are the prominent sound producers; however, in the Atlantic croaker and black drum, both male and female have sonic muscles and call (*Hill, Michael & Musick, 1987*; *Tellechea et al., 2010*). In most cases, sound production is associated with defending territory and reproduction. However, there are still some questions on how exactly calling and spawning are correlated in many sound-producing species (e.g., *Luczkovich et al., 1999a*; *Locascio, Burghart & Mann, 2012*).

Research studies have illustrated that patterns of fish calling overlap with patterns of reproductive senescence in the wild (*Connaughton & Taylor, 1995*). By performing acoustic recordings and plankton tows simultaneously, studies have also shown an association between calling and spawning (*Mok & Gilmore, 1983*; *Saucier & Baltz, 1993*; *Luczkovich et al., 1999a*; *Aalbers & Drawbridge, 2008*). These comparisons are necessary if acoustic metrics are to be used as a means to monitor fish spawning patterns. These data are difficult to acquire in the wild because of the uncertainties associated with ensuring that the collected eggs are from the same fish that are calling (*Locascio, Burghart & Mann, 2012*). In the field, it is likely that egg counts are affected by water currents, predator activity, and plankton tow efficiency. Captive studies can remove these variables. A few studies have mimicked wild environmental conditions and examined the relationship between calling and spawning in the laboratory (*Guest & Lasswell, 1978*; *Connaughton & Taylor,*

*1996*; *Montie et al., 2016*). Recently, *Montie et al. (2016)* collected quantitative data to better understand the association between sound production, call structure, and eggs released in a captive brood stock of red drum. In this study, red drum spawned only on evenings when males called. In addition, more eggs were collected on evenings when red drum calls were longer in duration and contained more pulses.

In the present study, our main objective was to examine the relationship between acoustic metrics (i.e., the amount of calling and sound pressure levels) and the number of eggs collected in laboratory tanks containing spotted seatrout. Spotted seatrout are found in estuaries from Cape Cod, Massachusetts to Key West, Florida and from southwest Florida to southern Mexico in the Gulf of Campeche (*Welsh & Breder jr, 1924*; *Mather, 1952*; *Tabb, 1966*). Spotted seatrout are group-synchronous spawners with a long spawning season from April to September along the South Atlantic and the Gulf of Mexico coasts (*Overstreet, 1983*; *Brown-Peterson, Thomas & Arnold, 1988*; *McMichael & Peters, 1989*; *Saucier & Baltz, 1993*; *Brown-Peterson et al., 2001*; *Brown-Peterson & Warren, 2002*; *Nieland, Thomas & Wilson, 2002*; *Brown-Peterson, 2003*; *Roumillat & Brouwer, 2004*). It was estimated that spotted seatrout in age classes 1–3 spawn every 4.7, 4.2, and 4 days, respectively, in Charleston Harbor, South Carolina (*Roumillat & Brouwer, 2004*). *Tabb (1966)* was the first to report that male spotted seatrout use acoustic signals that are associated with reproductive activity. *Mok & Gilmore (1983)* discovered that male spotted seatrout produce four major sound types, which were all recorded at spawning sites in Indian River Lagoon, Florida. These calls were designated as (i) a "grunt" followed by a series of "knocks", (ii) "aggregated grunts," (iii) "long grunt"; and (iv) a rapid series of short pulses known as the "staccato." Females do not have a sonic muscle and do not produce sound (*Mok & Gilmore, 1983*). It is well known that spotted seatrout form large mating aggregations (i.e., leks). These aggregations produce higher sound pressure levels (>50 dB–90 dB re 1 μPa at 3 m) as compared to the levels of individual callers (<30 dB re 1 μPa at 3 m) (*Gilmore, 2003*). It is speculated that this combined acoustic energy derived from a male aggregation increases the number of receptive spawning females attracted to the lek (*Gilmore, 2003*). In the present study, our specific objectives were to: (i) briefly describe the different call types of captive spotted seatrout; (ii) investigate if captive fish showed daily rhythms in calling; (iii) determine if a quantitative relationship exists between the amount of calling and the quantity of eggs released; and (iv) examine if certain call types or changes in call structure were associated with spawning productivity. These data provide supportive evidence that acoustic metrics can be used to better understand reproductive rhythms over various time scales (i.e., daily, tidal, lunar, seasonal, yearly, and decadal) and that these metrics can be used to investigate the impacts of natural and anthropogenic stressors on these reproductive patterns.

## MATERIALS AND METHODS

For this study, sexually mature spotted seatrout were placed into three separate, 3.67 m diameter, 1.7 m deep, circular fiberglass tanks (i.e., Tank 1, Tank 2, and Tank 3) (Fig. 1A). Each tank was filled with settled, sterilized Charleston Harbor seawater and contained individual re-circulating aquaculture systems outfitted with UV sterilizers,

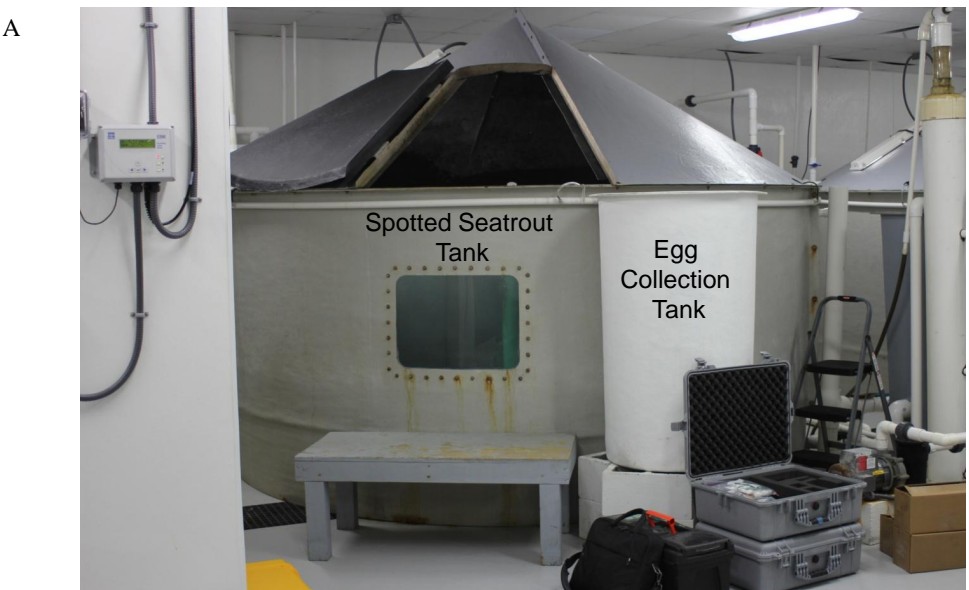

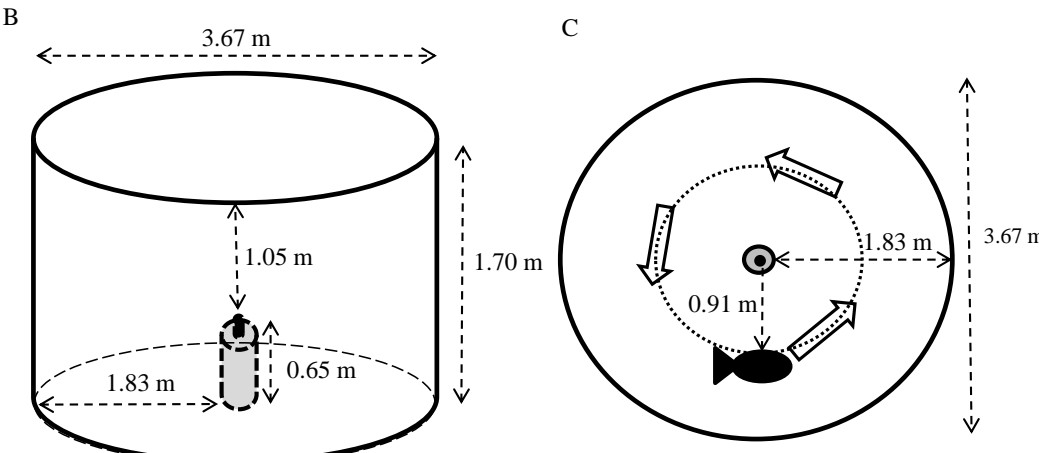

**Figure 1** **Tank setup.** (A) Spotted seatrout (*Cynoscion nebulosus*) were held in three separate, 3.67 m diameter fiberglass tanks (i.e., Tank 1, Tank 2, and Tank 3) with individual recirculating aquaculture systems equipped with UV sterilizers, protein fractionators, and bead filters. Floating eggs were collected from a surface, skimming port in the side of the tank that drained into an egg collection tank equipped with a 250-micron mesh net. (B) Placement of DSG-Ocean acoustic recorders relative to the tank's dimensions. (C) Placement of DSG-Oceans relative to the general swimming patterns of the fish.

protein fractionators, and bead filters. Fish numbers, sexes, and sizes for each tank are provided in Table 1. Tank 1 housed seven males and seven females. Tank 2 contained eight males and seven females. Tank 3 had three males and thirteen females. Three times a week, fish were fed equal parts of shrimp, squid, and Boston mackerel (*Scomber scombrus*). Tank temperatures were individually controlled. Water temperature and photoperiod were held at a cycle that encouraged spawning and followed a natural reproductive season for spotted

**Table 1 Summary of information for spotted seatrout (*Cynoscion nebulosus*) in each tank.**

| Tank information | Tank 1 | Tank 2 | Tank 3 | Means ± SE |
|---|---|---|---|---|
| No. of males | 7 | 8 | 3 | 6 ± 2 |
| Mean weight of males (g) | 1,126 ± 320 | 1,314 ± 281 | 1,200 ± 194 | 1,213 ± 55 |
| Mean length of males (mm) | 469 ± 41 | 500 ± 36 | 484 ± 29 | 484 ± 9 |
| No. of females | 7 | 7 | 13 | 9 ± 2 |
| Mean weight of females (g) | 1,298 ± 194 | 1,953 ± 670 | 1,403 ± 209 | 1,551 ± 203 |
| Mean length of females (mm) | 497 ± 29 | 543 ± 70 | 502 ± 28 | 514 ± 15 |
| Timeframe for data collection | 4/13/12–12/19/12 | 4/13/12–11/21/12 | 4/13/12–11/21/12 | NA |
| No. of days monitored | 250 | 222 | 222 | 231 ± 9 |
| Mean water temperature (°C) | 26.4 | 24.9 | 26.0 | 25.8 ± 0.4 |

**Notes.**
NA, not applicable.
Means ± standard deviations for individual tanks.
Means ± standard errors of all four tanks.

seatrout in South Carolina (*Roumillat & Brouwer, 2004*; *Montie, Vega & Powell, 2015*). However, in Tank 1, the temperature was maintained at 27.8 °C and the photoperiod was left at 14.5 h light until late November to extend the spawning season and provide more data points to test the association between calling and spawning. Acoustic monitoring occurred from April 13 to December 19, 2012 for Tank 1 and from April 13 to November 21, 2012 for Tanks 2 and 3. Different fish compositions, temperature cycles, and photoperiods among the tanks were set by SCDNR to investigate how these variables affected reproductive success.

In order to collect floating eggs from spawns, the large tanks contained a surface, skimming port that emptied into an egg collection tank following methods previously described (*Montie et al., 2016*). Each morning, the nets were checked for eggs. If present, the eggs were harvested and transferred to a 15 L *Artemia* hatching cone, which allowed separation into fertile (i.e., floating) and non-fertile (i.e., sinking) eggs. Once separated, fertile and non-fertile eggs were enumerated volumetrically using graduated cylinders. In all results reported in this study, the quantity of eggs collected refers to the sum of the fertile and non-fertile eggs. This work was part of the SCDNR restocking program for spotted seatrout and did not require an IACUC. No fish were harmed or sacrificed in this study.

We deployed acoustic loggers (DSG-Oceans, Loggerhead Instruments, Sarasota, FL, USA, www.loggerhead.com) into each tank following methods previously described (*Montie et al., 2016*). The DSG-Ocean was positioned vertically in the tank center (i.e., 3.67 diameter tanks) inside a cement mold and attached to a piece of vertical rebar, which was embedded in the cement mold (Fig. 1B). The DSG-Ocean was approximately 1.83 m from the side of the tank and 1.05 m from the water surface. Observations of fish in tanks indicated that the average swimming patterns occurred counterclockwise at mid-depth (i.e., 0.85 m) approximately half way between the hydrophone and the tank side (Fig. 1C).

When recording fish sounds in tanks, there is always the possibility that reverberation, resonance, and tank size can affect the sounds recorded. For example, *Parmentier et al. (2014)* provided some evidence that fiberglass tanks (i.e., 6 m³ and 13 m³) can affect

sound duration, pulse duration, pulse period, and dominant frequency. *Akamatsu et al. (2002)* illustrated that dominant frequency, sound pressure level, and power spectrum recorded in very small, 170 L (i.e., 0.17 m$^3$) rectangular glass tanks were significantly distorted compared to the original acoustic signal. Nonetheless, most captive studies that focus on sound production live with resonance and distortion effects present in tanks (e.g., *Connaughton & Taylor, 1996*; *Montie et al., 2016*). *Akamatsu et al. (2002)* do provide a practical procedure for correcting the measurements of fish sounds in small tanks. These guidelines indicate that the targeted sound frequency (i.e., the frequency of the fish calls) should be different than the minimum resonant frequency ($f_{min}$) of the tank. In addition, the hydrophone should be positioned within the attenuation distance (D) of the sound source to minimize possible distortion.

Even though *Akamatsu et al. (2002)* used tanks that were much smaller than the tanks used in the present study (i.e., 0.17 compared to 17.88 m$^3$), we calculated $F_{min}$ and D using the guidelines and equations provided by *Akamatsu et al. (2002)*. We determined that $F_{min} = 541$ Hz, which was different than the expected peak frequency of spotted seatrout chorusing in the wild (i.e., ~239 Hz; E Montie, 2016, unpublished data); thus possible resonance was minimal. We also determined that D = 1.48 m, which was smaller than the radius of the tank and the maximum possible distance of a calling fish from the hydrophone (i.e., 1.83 m); thus some distortion was possible. We minimized this distortion effect by placing the recorders in the center of the tank. Therefore, calling fish ranged anywhere from 0 m (i.e., when a fish was immediately next to recorder) to 1.83 m (i.e., when a fish was located next to the side of the tank). As stated previously, observations of fish in these tanks indicated that the average swimming patterns occurred counterclockwise half way between the hydrophone and the tank side (i.e., ~1 m from the hydrophone). Therefore, generally speaking, calling fish were less than 1 m from the hydrophone, which was within the attenuation distance of the sound source and distortion was minimized.

The DSG-Ocean contains a High Tech Inc. hydrophone (i.e., −185 dBV μPa$^{-1}$ sensitivity) attached to a microcomputer circuit board with a gain of 20 dB that is powered by 24 D-cell alkaline batteries. This equipment is housed in a cylindrical PVC housing (i.e., 0.65 cm length, 11.5 cm diameter). The DSG board is calibrated with a 0.1 V (peak) frequency sweep from 2–100 kHz. For this experiment, DSG-Oceans were set to a sampling rate of 50 kHz and were scheduled to record sound for 2 min every 20 min. Files were saved as 'DSG files' on a 128 GB SD-card. Recorders were retrieved nine times during the experiment, once on May 5th, May 7th, May 25th, June 25th, July 23rd, August 24th, September 21st, and October 26th to download data and change batteries and again on November 21st and December 19th to download final files. In Tank 1, acoustic recordings were not collected between 16:40, May 18th and 19:40, May 22nd, 2012 due to recorder malfunction. After each data retrieval, the 'DSG files' were downloaded to a network drive and batch converted into 'wav files' using DSG2wav©software (Loggerhead Instruments, Sarasota, FL, USA).

We manually counted the quantity of calls within each 'wav file' by viewing the files in Adobe Audition (Adobe Systems Incorporated, San Jose, CA, USA, http://www.adobe.com). In our captive study, we grouped the ''grunt followed by knocks''

and "aggregated grunts" observed by *Mok & Gilmore (1983)* together because of their similarity in call structure and described these calls as "drums" to avoid confusion with the "grunt." We classified the "long grunt" as the "grunt," while the "staccato" strictly followed the description provided by *Mok & Gilmore (1983)*. For each tank, we determined the number of "grunts," "drums," and "staccatos" per day by adding up the calls that occurred between 18:00 and 06:00, which was the time range in which most calling occurred. The 'wav' file with the most abundant calls was used to determine the mean number of pulses in a "staccato" call as well as the mean duration for that specific day. The pulse number was evaluated by manually counting each individual pulse in a "staccato." The "staccato" duration was determined by manually subtracting the time of call termination from the time of call initiation.

We determined the received root mean square (rms) sound pressure level (SPL; dB re 1 uPa; between 50 and 2,000 Hz) of each 'wav file' using MATLAB (The MathWorks, Inc., Natick, MA, USA, http://www.mathworks.com). Noise due to aeration, pumps, filters, and electrical systems between 50 and 2,000 Hz was minimal compared to the SPL of calling spotted seatrout (see Figs. 2 and 3). The equations in our MATLAB script for received SPL determination followed PAMGuide scripts as described by *Merchant et al. (2015)*:

$$S = h + g + 20 \log 10(1/V_{adc});$$
$$b = 20 \log 10(\mathrm{sqrt}(\mathrm{mean}(y^2)));$$
$$a = b - S;$$

where $a$ = calibrated sound level in dB re 1 μPa; $b$ = uncorrected signal; $S$ = correction factor; $h$ = hydrophone sensitivity (i.e., $-185$ dBV μPa$^{-1}$) ; $g$ = DSG gain (i.e., 20); $V_{adc}$ = analog-to-digital conversion (i.e., 1 volt); $y$ = signal. Wav files that had noise artifacts due to fish hitting the tank or recorder and tank maintenance were not included in SPL analysis. From these data, we then calculated the mean rms SPL per day for each tank by calculating the mean of all the SPLs between 18:00 and 06:00. For Tanks 1 through 3, the average background noise levels were 114, 113, and 112 dB re 1 μPa, while the highest received rms SPLs when spotted seatrout were calling were 148 (i.e., 139 calls detected), 146 (i.e., 76 calls detected), and 138 (i.e., 67 calls detected) dB re 1 μPa, respectively.

We used Microsoft Excel (Microsoft, Redmond, WA, USA; http://www.microsoft.com/en-us), MATLAB, and SYSTAT 13 (Systat Software, Inc., San Jose, CA, USA; http://www.systat.com) for data and statistical analysis. First, the time domain and power spectral density (PSD) were illustrated for "drums," "grunts," and "staccatos." Then, the number of calls and received SPLs, spawning productivity, and call characteristics for each tank were summarized. Using Pearson correlation analysis, we investigated the relationship between all calls (i.e., sum of "grunts," "drums," and "staccatos") and received SPLs for each tank. To examine the relationship between simulated seasonal parameters and calling, we plotted water temperature, photoperiod, and the number of calls versus date. To investigate the daily rhythms of calling, we determined the mean number of "drums" for each time interval during the 14.5 h of light photoperiod for each tank.

We performed a quantitative, stepwise investigation to determine the relationship between calling and spawning. In order to examine the overlap of sound production and

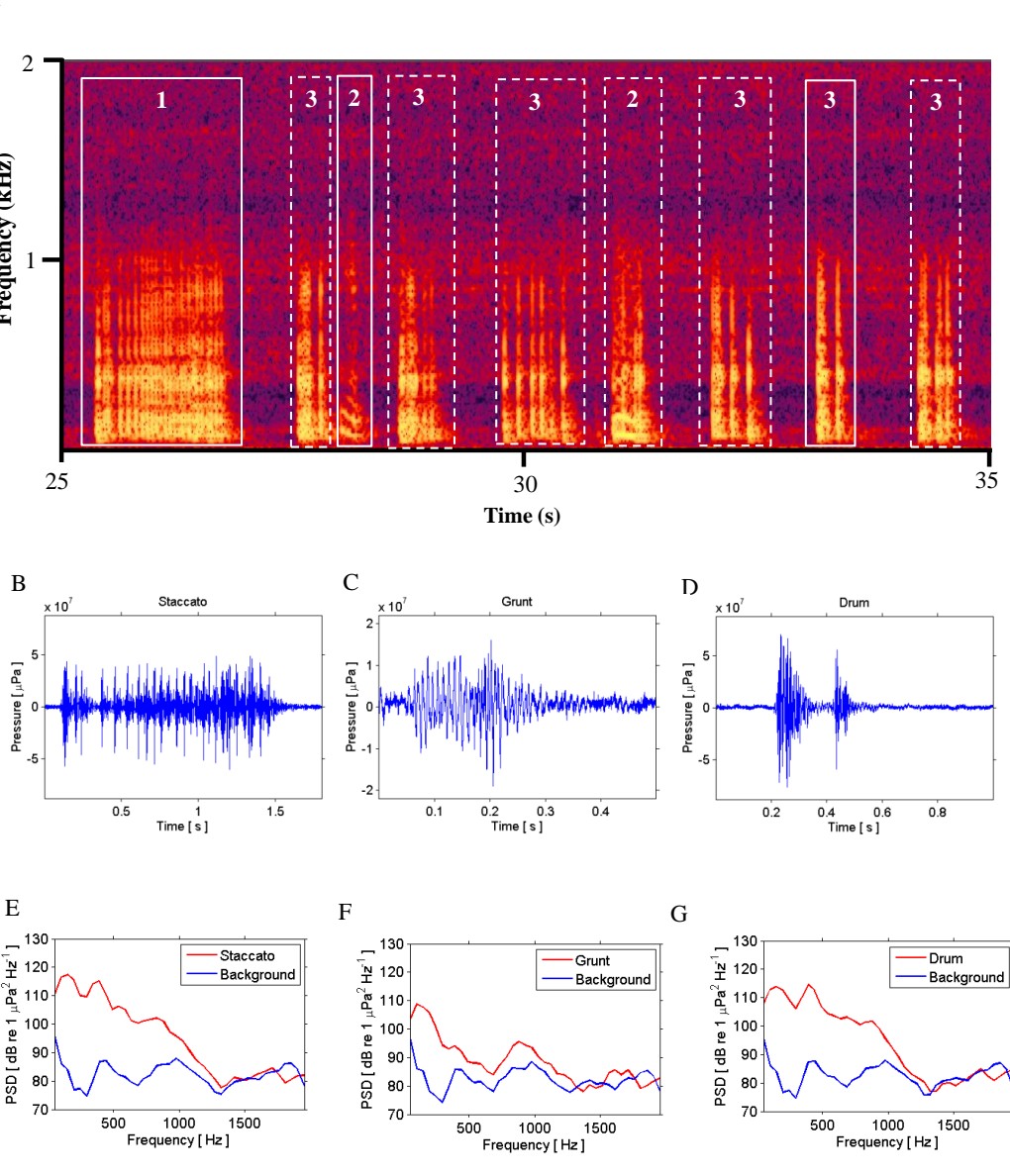

**Figure 2 Call Types.** Different call types produced by wild caught spotted seatrout (*Cynoscion nebulosus*) held in captivity. Spotted seatrout produced three different call types. These calls were characterized as "drums," "grunts," or "staccatos" following similar nomenclature published in other studies (*Mok & Gilmore, 1983*; *Sprague et al., 2000*; *Walters et al., 2009*). (A) A continuous spectrogram illustrating a staccato (labeled 1), grunts (labeled 2), and a series of drums (labeled 3). Time domain of (B) a staccato, (C) a grunt, and (D) a drum call. Power spectral density (PSD) plots of (E) a staccato, (F) a grunt, and (G) a drum call. The sample rate was 50 kHz. In (A) the spectrogram was created using Adobe Audition with a spectral resolution of 2,048 and time duration of 10 s. Brighter colors correspond to higher sound pressure levels. Time domain and PSD figures correspond to the calls outlined in solid white lines in (A). PSDs of background noise in the tanks were calculated from recordings when fish were not present in the tanks. PSDs were determined using a FFT size of 1,024 samples, which corresponded to a frequency resolution of 48.8 Hz.
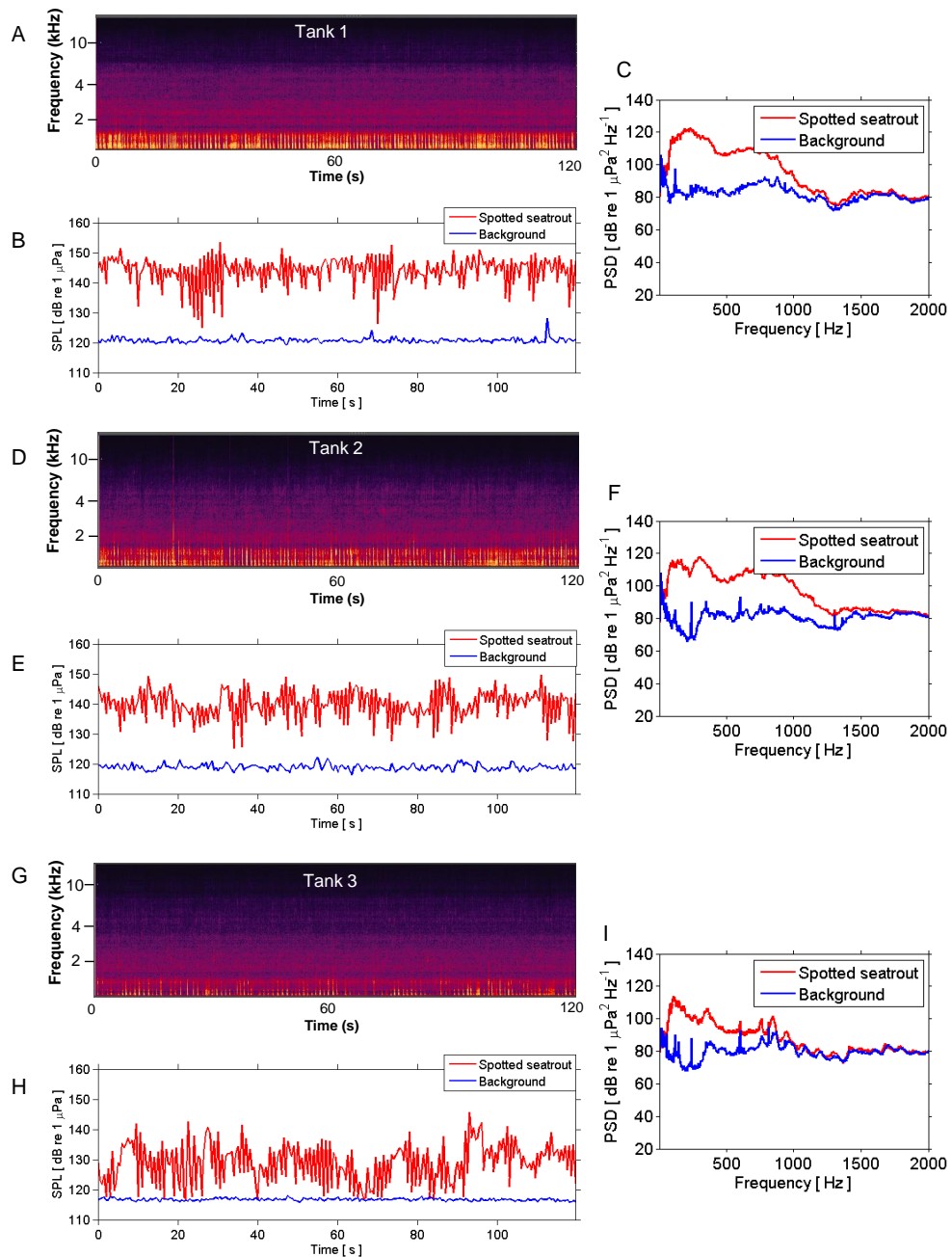

**Figure 3** **SPL and PSD comparisons between background and spotted seatrout choruses.** Representa-
tive spectrograms (A, D, G), instantaneous sound pressure levels (B, E, H), and power spectral density cal-
culations (C, F, I) of an entire 2 min wav file for each tank during an evening in which male spotted seat
trout called and females spawned. Tank 1 = 2,680,000 eggs were collected the next morning; Tank 2 =
7,110,000 eggs; and Tank 3 = 2,580,000 eggs. The sample rate was 50 kHz. The spectrograms were created
using Adobe Audition with a spectral resolution of 2,048 and time duration of 120 s. Brighter colors cor-
respond to fish calling and higher sound pressure levels. Instantaneous SPLs and PSDs in this figure were
calculated between 1 and 2,000 Hz using a FFT size of 50,000 samples, which corresponded to a frequency
resolution of 1 Hz. Corresponding background SPLs and PSDs for each tank were performed on 2 min
wav files in which calling was not detected. These background noise files were selected as closely in time as
possible to the represented fish choruses.

reproductive events, our first approach was to plot the total number of calls per day and the quantity of eggs collected (i.e., the next morning) versus the date for each tank. The next step was to perform autocorrelation analysis (i.e., at −5 to +5 day lags) to examine the relationship between calling variables (i.e., number of calls and received SPL) and eggs collected. This analysis determined if the correlation between calling and spawning was the highest when these two variables were aligned within the same timeframe as compared to a negative or positive lag. Then, we used logistic regression to test whether or not the likelihood that spotted seatrout tended to spawn increased as calling (i.e., the total quantity of calls between 15:40 and 00:00) and received SPLs (i.e., the mean SPL between 15:40 and 00:00) increased. The categorical dependent variable was spawning (0 = no; 1 = yes) and the predictor variables were calling or received SPL. Next, we used linear regression analysis to investigate if more productive spawns were associated with more calling or higher sound levels. We performed two separate linear regressions. One regression analysis included all the data, and the other analysis excluded the days when there were no eggs. We also performed logistic regression to test whether or not the likelihood that spotted seatrout tended to spawn increased as "staccato" calls increased in duration and pulse number. To test the relationship between call structure and spawning productivity, we performed linear regression analysis with call duration or pulse number as the independent variable and the quantity of eggs as the dependent variable.

## RESULTS

### Types of spotted seatrout calls detected in captivity

Spotted seatrout produced three call types. These calls were characterized as "drums," "grunts," or "staccatos" (Fig. 2; *Mok & Gilmore, 1983*; *Sprague et al., 2000*; *Walters et al., 2009*). We provided a very coarse description of these call types and leave a more detailed description of frequency range, peak frequency, sound pressure level, call duration, number of pulses in a call, pulse duration, and pulse interval for each of the call types to future investigations. What we report here is that most of the acoustic energy of calling occurred between 50 and 1,000 Hz (Figs. 2E–2G). A staccato call was characterized as having multiple pulses ($n > 5$) (Figs. 2A, 2B, 2E). A grunt call was composed of a single pulse displaying multiple harmonics (Figs. 2A, 2C, 2F). A drum call was composed of one to five pulses (Figs. 2A, 2D, 2G). In all tanks, the number of drums, grunts, and staccatos were positively correlated with each other (Pearson Correlation Test; $P < 0.05$ for all comparisons). Sound pressure levels during periods when fish were calling were higher than the background noise levels when spotted seatrout were not calling (Figs. 3B, 3E, 3H). In all tanks, drums were the most frequently produced followed by grunts and then staccatos (Table 2; Fig. 4). Sound pressure level and the total number of calls (i.e., sum of drums, grunts, and staccatos) were positively correlated (Pearson Correlation Test; $r = 0.917$, $P < 0.01$ for Tank 1; $r = 0.688$, $P < 0.01$ for Tank 2; and $r = 0.457$, $P < 0.01$ for Tank 3).

### Seasonal and daily patterns in calling

Three general patterns in calling were observed over the simulated reproductive season. First, fish calling occurred in all tanks (Table 2; Fig. 4). Second, the amount of calling

**Table 2  Tank summary of sound production and spawning events for captive spotted seatrout (*Cynoscion nebulosus*).**

| Tank Information | Tank 1 | Tank 2 | Tank 3 | Means ± SE |
|---|---|---|---|---|
| No. of spawns | 81 | 13 | 3 | 32 ± 25 |
| No. of spawns/days monitored | 0.32 | 0.06 | 0.01 | 0.13 ± 0.10 |
| Eggs collected | 72,486,000 | 13,630,000 | 4,160,000 | 30,092,000 ± 21,372,558 |
| Eggs collected/days monitored | 289,944 | 61,396 | 18,739 | 123,360 ± 84,198 |
| No. of drums | 227,659 | 123,729 | 15,532 | 122,307 ± 61,240 |
| No. of drums/days monitored | 911 | 557 | 70 | 513 ± 244 |
| Mean drums between 18:00 to 06:00 (no spawning) | 708 ± 516 | 491 ± 467 | 59 ± 123 | 419 ± 191 |
| Mean drums between 18:00 to 06:00 (spawning) | 1,376 ± 538 | 1,624 ± 325 | 856 ± 442 | 1,285 ± 226 |
| No. of grunts | 13,109 | 6,105 | 1,786 | 7,000 ± 3,299 |
| No. of grunts / days monitored | 52 | 28 | 8 | 29 ± 13 |
| Mean grunts between 18:00 to 06:00 (no spawning) | 41 ± 32 | 25 ± 24 | 7 ± 15 | 24 ± 10 |
| Mean grunts between 18:00 to 06:00 (spawning) | 78 ± 45 | 69 ± 28 | 63 ± 51 | 70 ± 4 |
| No. of staccatos | 3,139 | 1,565 | 22 | 1,575 ± 900 |
| No. of staccatos / days monitored | 13 | 7 | <1 | 10 ± 2 |
| Mean staccatos between 18:00 to 06:00 (no spawning) | 8 ± 8 | 6 ± 10 | <1 | 7 ± 1 |
| Mean staccatos between 18:00 to 06:00 (spawning) | 24 ± 14 | 29 ± 20 | 3 ± 4 | 19 ± 8 |
| Mean SPL between 18:00 to 06:00 (no spawning) | 120 ± 3 | 116 ± 3 | 113 ± 2 | 116 ± 2 |
| Mean SPL between 18:00 to 06:00 (spawning) | 124 ± 3 | 122 ± 2 | 118 ± 4 | 121 ± 2 |

**Notes.**

SPL, received sound pressure level (dB re 1 uPa).
Means ± standard deviations reported for individual tanks.
Means ± standard errors of all four tanks.

differed among tanks (Tank 1 > Tank 2 > Tank 3; Table 2; Fig. 4). Third, sound production changed with photoperiod and water temperature adjustments. Maximal calling occurred when the photoperiod shifted to 14.5 h of light, and the temperature increased to 27.7 °C. Sound production began to decrease as the temperature dropped below 27.7 °C, and the light cycle changed to 12 h light per day. In Tank 3, spotted seatrout calling was more sporadic (Fig. 4C). The general pattern was that calling increased as the light cycle shifted from 12.5 to 14.5 h light per day and as the temperature increased to 27.8 °C. Between 9/21/2012 and 11/8/2012, calling was more prevalent than other time periods. Sound production began to diminish as the temperature dropped below 24.0 °C on 11/14/2012. In Tanks 1–3, rapid fluctuations in water temperature affected sound production; rapid elevations in temperature increased calling rates, while rapid declines in temperature decreased calling (Fig. 4).

In all tanks, spotted seatrout showed daily rhythms in sound production (Fig. 5). Generally, calling began once the lights turned off (i.e., 17:45). The highest number of drums occurred at 21:20 in Tank 1; 21:00 in Tank 2; and 20:40 in Tank 3 (Fig. 5). The number of grunts, staccatos, and received SPLs followed similar patterns.

## The Association between Calling and Spawning

In this captive experiment, our data indicated that sound production served an important function in courtship. First, we discovered that successful reproductive events happened only on evenings in which spotted seatrout produced sound (Fig. 6). On many evenings,

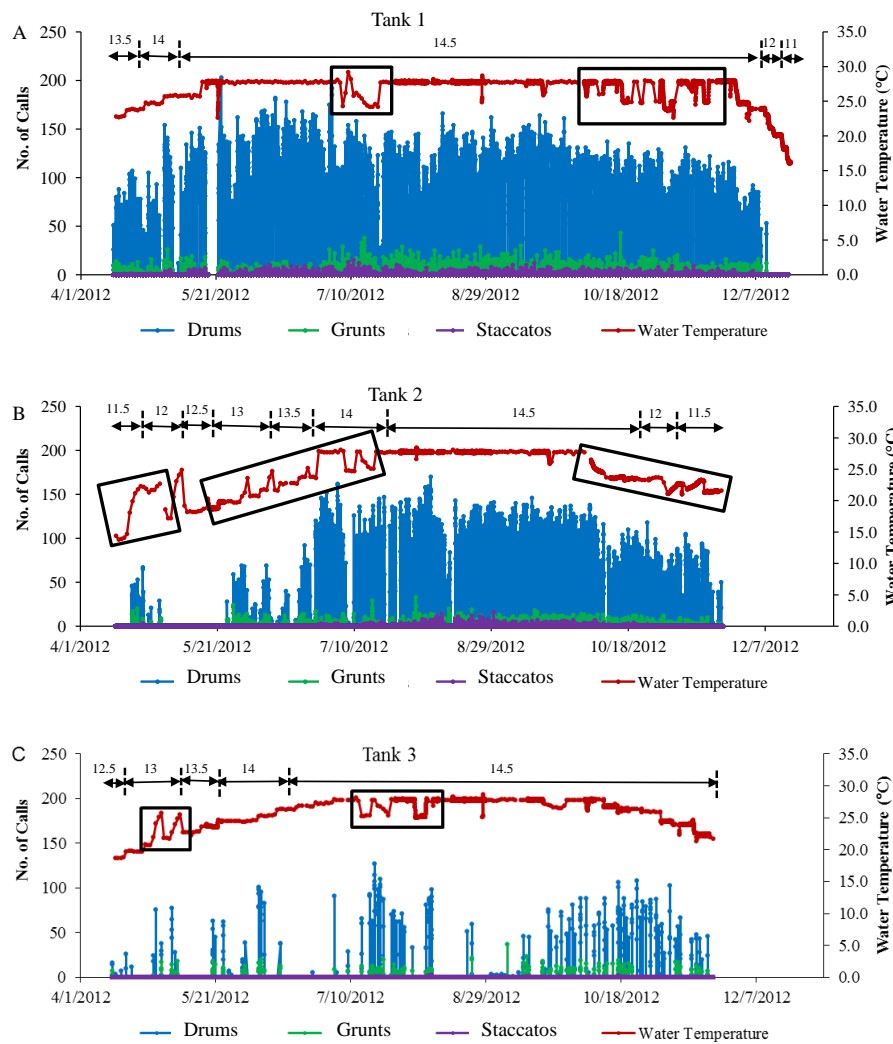

**Figure 4  Sound production of spotted seatrout throughout study period.** Sound production by wild caught spotted seatrout (*Cynoscion nebulosus*) held in captivity throughout the entire study period. The number of drums, grunts, and staccatos in each 2 min 'wav file' was manually counted by an observer and plotted versus date with corresponding water temperatures for (A) Tank 1, (B) Tank 2, and (C) Tank 3. The numbers above the horizontal arrows indicate the number of hours of light present in the respective photoperiod. Boxes indicate rapid fluctuations in water temperature. Generally, abrupt rises in temperature were followed by an increase in calling, while abrupt drops were followed by a decrease in the amount of calling.

male spotted seatrout did call and no spawning was observed, but spawning never happened without a significant increase in sound production. Second, autocorrelation analysis showed that the greatest correlation between sound production and eggs released happened on the same evening when the lag = 0 (Fig. 7). Third, logistic regression results indicated that the likelihood that spotted seatrout reproduced was significantly related to calling variables (i.e., number of calls and received SPL) (Table 3; Fig. 8). Fourth, there was some evidence that spawning was more productive when spotted seatrout called more frequently. When all data were included in regression analysis, more productive spawns (i.e., larger egg

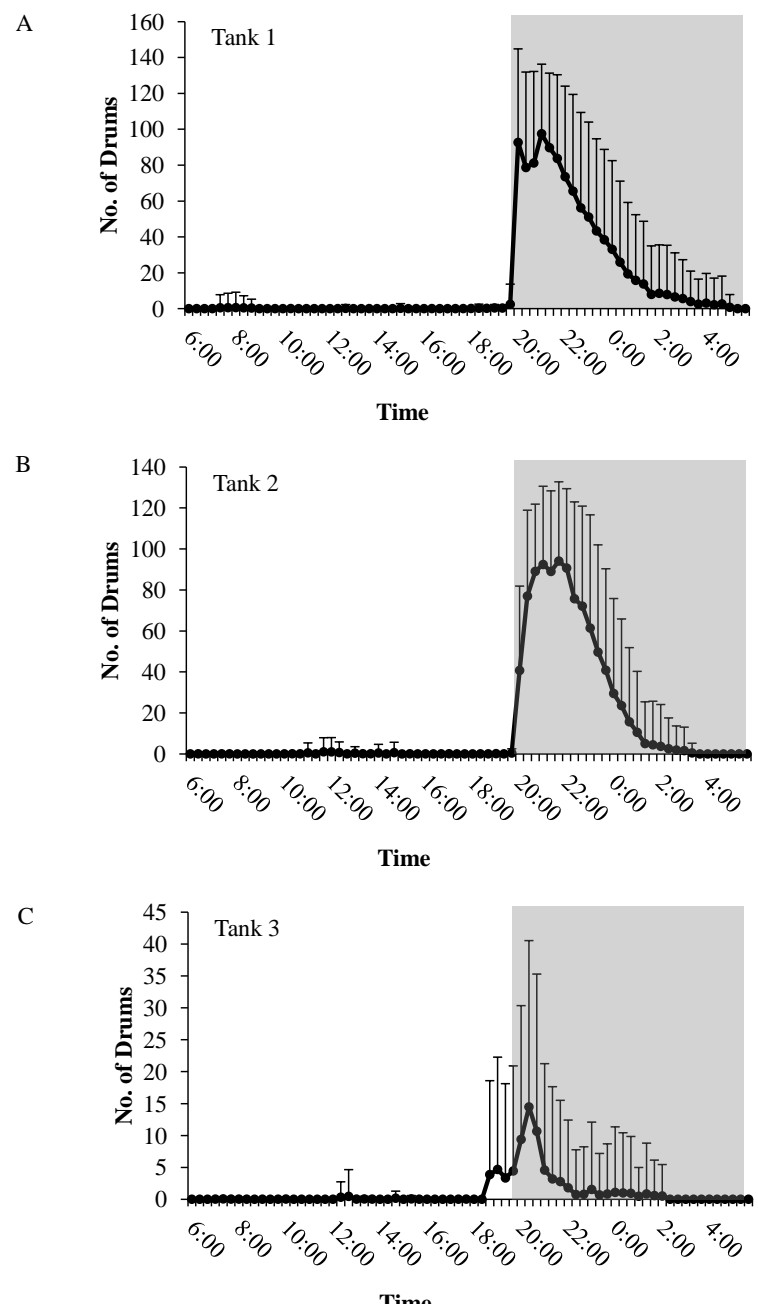

**Figure 5 Daily patterns of sound production.** Daily patterns of sound production by spotted seatrout (*Cynoscion nebulosus*) in (A) Tank 1; (B) Tank 2; and (C) Tank 3. To examine these patterns, we determined the mean number of drums for each time interval (e.g., 12:00–12:02; 12:20–12:22; 12:40–12:42, 13:00–13:02, etc.) during the 14.5 h light photoperiod. The grey box indicates the time span of darkness during the 14.5 h light photoperiod. Standard deviations are reported as vertical bars.

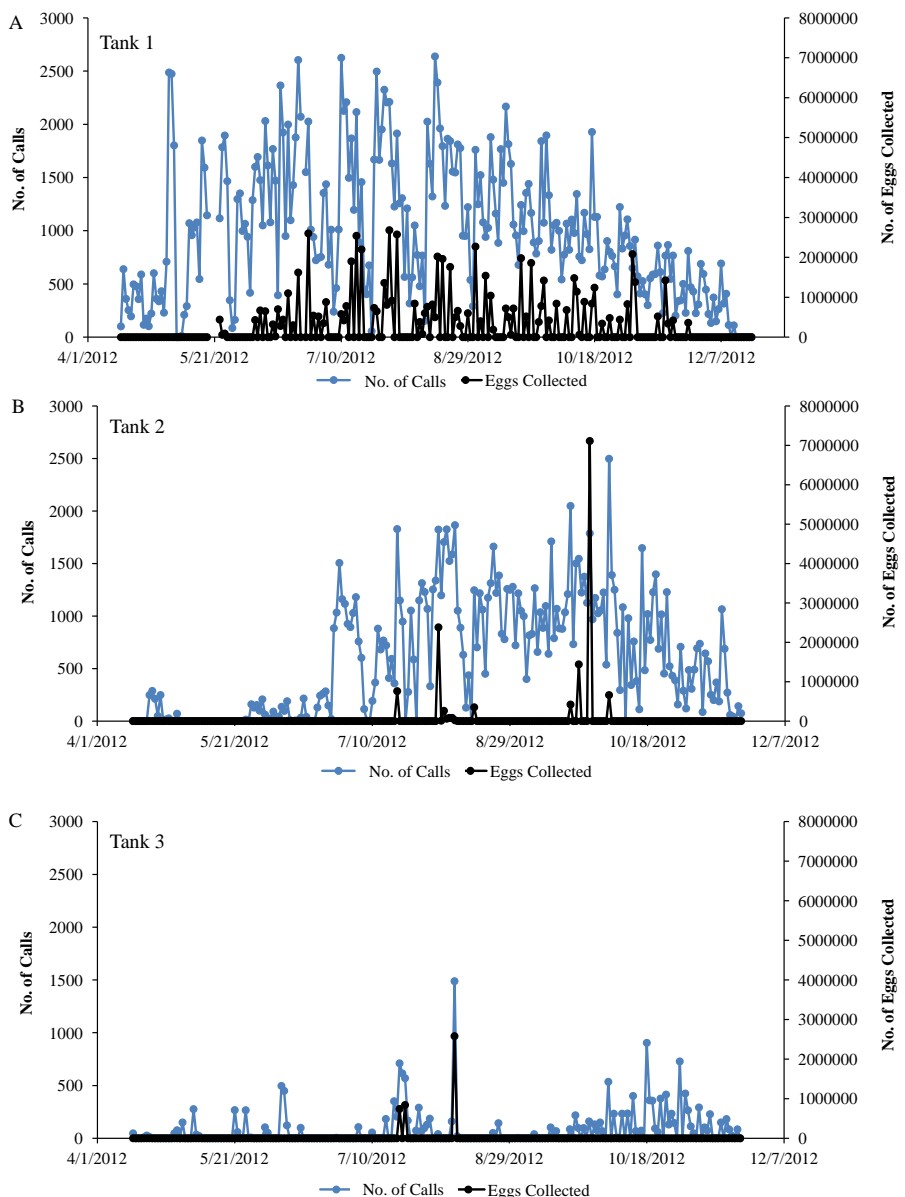

**Figure 6 Sound production and spawning.** Sound production and spawning of wild caught spotted seatrout (*Cynoscion nebulosus*) held in captivity throughout the entire study period. Calls per day and the number of eggs collected (i.e., the next morning) were plotted versus the date for (A) Tank 1, (B) Tank 2, and (C) Tank 3.

numbers) were associated with more calling and higher received SPLs (Table 4). If the days when no eggs were collected were removed from statistical tests, then the amount of calling in relation to egg deposition was not significant (Table 4). However, in Tanks 1 and 3, the relationships between SPL and the number of eggs collected were significant, even when the non-spawning events were removed from linear regressions (Table 4). Fifth, tanks with more sound production had more spawns (Tank 1 > Tank 2 > Tank 3; Table 1). Tanks with more calling and higher mean SPLs resulted in larger total egg yields per gram female

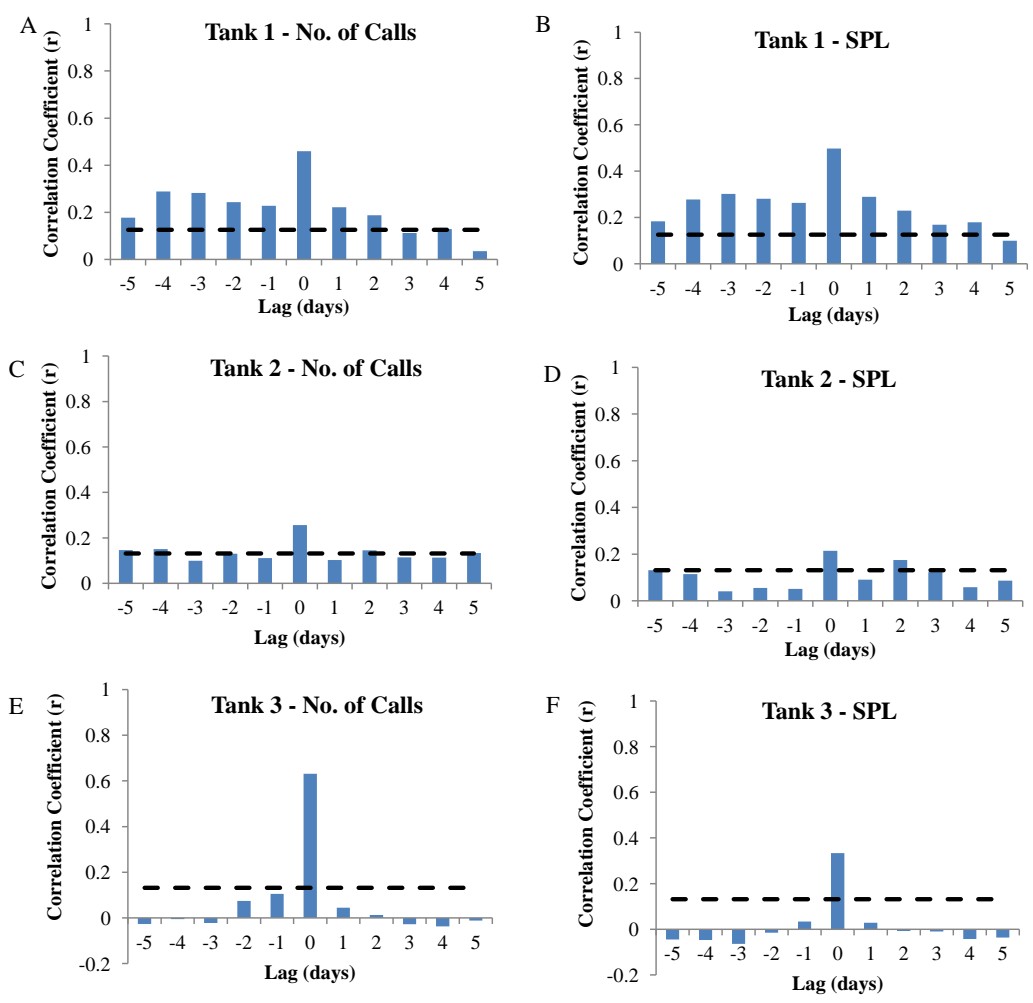

**Figure 7** **Autocorrelation analysis.** Autocorrelation analysis was used to determine the correlation between calling variables (i.e., number of calls and sound pressure level or SPL) and eggs collected at −5 to +5 day lags in wild caught spotted seatrout (*Cynoscion nebulosus*) held in captivity. (A) Tank 1 –number of calls, (B) Tank 1 –SPL, (C) Tank 2 –number of calls, (D) Tank 2 –SPL, (E) Tank 3 –number of calls, (F) Tank 3 –SPL. The dotted lines represent the 95% confidence interval.

biomass (Fig. 9). We did test whether or not spotted seatrout were more likely to spawn when staccato calls contained more pulses and calls were longer in duration, but we found no significant difference.

## DISCUSSION

### Spotted seatrout call types in captivity

Spotted seatrout call types in this study were similar to the types of calls described in other studies (*Mok & Gilmore, 1983*; *Luczkovich et al., 1999b*; *Sprague et al., 2000*; *Luczkovich et al., 2008*). *Mok & Gilmore (1983)* recorded and classified spotted seatrout sounds into four call types: (i) "grunt followed by knocks"; (ii) "aggregated grunts"; (iii) "long grunt"; and (iv) "staccato." In wild studies, these four call types have been recorded at spawning

**Table 3** **Results of logistic regression analysis that tested whether or not the amount of calling and sound pressure level (SPL) were significant predictors of spotted seatrout (*Cynoscion nebulosus*) spawning.** In all cases, the dependent variable was spawning (0 = no; 1 = yes). No. of calls and mean SPL are measurements per evening from 15:40 to 0:00. The overall model evaluation used the log-likelihood statistical test.

| Tank | Predictor variable | Logistic model | $z$ | $P$ | Odds ratio ($e^{\beta}$) | Overall model evaluation | | |
|---|---|---|---|---|---|---|---|---|
| | | | | | | $X^2$ | $df$ | $P$ |
| Tank 1 | No. of drums | $Logit(y) = -2.926 + 0.002x$ | 7.073 | <0.001 | 1.002 | 70.216 | 1 | <0.001 |
| Tank 1 | No. of grunts | $Logit(y) = -2.154 + 0.026x$ | 5.873 | <0.001 | 1.026 | 46.545 | 1 | <0.001 |
| Tank 1 | No. of staccatos | $Logit(y) = -2.555 + 0.132x$ | 7.495 | <0.001 | 1.141 | 94.895 | 1 | <0.001 |
| Tank 1 | Total calls | $Logit(y) = -2.983 + 0.002x$ | 7.123 | <0.001 | 1.002 | 72.215 | 1 | <0.001 |
| Tank 1 | Mean SPL | $Logit(y) = -42.521 + 0.344x$ | 6.793 | <0.001 | 1.411 | 66.415 | 1 | <0.001 |
| Tank 2 | No. of drums | $Logit(y) = -12.522 + 0.008x$ | 4.172 | <0.001 | 1.008 | 64.594 | 1 | <0.001 |
| Tank 2 | No. of grunts | $Logit(y) = -5.902 + 0.066x$ | 4.130 | <0.001 | 1.068 | 31.143 | 1 | <0.001 |
| Tank 2 | No. of staccatos | $Logit(y) = -4.033 + 0.090x$ | 4.573 | <0.001 | 1.094 | 26.282 | 1 | <0.001 |
| Tank 2 | Total calls | $Logit(y) = -13.197 + 0.008x$ | 4.131 | <0.001 | 1.008 | 66.084 | 1 | <0.001 |
| Tank 2 | Mean SPL | $Logit(y) = -103.059 + 0.841x$ | 4.381 | <0.001 | 2.318 | 39.716 | 1 | <0.001 |
| Tank 3 | No. of drums | $Logit(y) = -7.385 + 0.010x$ | 2.784 | 0.005 | 1.010 | 20.540 | 1 | <0.001 |
| Tank 3 | No. of grunts | $Logit(y) = -5.653 + 0.055x$ | 3.257 | 0.001 | 1.057 | 11.028 | 1 | 0.001 |
| Tank 3 | No. of staccatos | $Logit(y) = -5.236 + 1.725x$ | 2.501 | 0.012 | 5.612 | 13.344 | 1 | <0.001 |
| Tank 3 | Total calls | $Logit(y) = -7.226 + 0.009x$ | 2.797 | 0.005 | 1.009 | 20.060 | 1 | <0.001 |
| Tank 3 | Mean SPL | $Logit(y) = -87.532 + 0.722x$ | 2.897 | 0.004 | 2.058 | 10.877 | 1 | 0.001 |

**Notes.**
SPL, received sound pressure level in dB re 1 $\mu$Pa.

**Table 4** **Results of linear regression analysis that tested the significance of the amount of calling and sound pressure level in relation to spawning success of spotted seatrout (*Cynoscion nebulosus*) held in captivity.** In all cases, the dependent variable is the number of eggs collected. $P$-values were statistically significant when $P < 0.050$.

| Tank | Independent variable | Fitted equation | $r^2$ | $P$ | $df$ |
|---|---|---|---|---|---|
| Tank 1 | Total no. of calls | $y = 402x - 104,522$ | 0.208 | <0.001 | 243 |
| Tank 1[a] | Total no. of calls[a] | $y = 246x + 530,897$[a] | 0.043[a] | 0.062[a] | 79[a] |
| Tank 1 | Mean SPL | $y = 74,752x - 8,745,163$ | 0.245 | <0.001 | 243 |
| Tank 1[a] | Mean SPL[a] | $y = 74,494x - 8,315,774$[a] | 0.136[a] | 0.001[a] | 79[a] |
| Tank 2 | Total no. of calls | $y = 235x - 77,917$ | 0.065 | <0.001 | 220 |
| Tank 2[a] | Total no. of calls[a] | $y = 714x - 180,228$[a] | 0.015[a] | 0.687[a] | 11[a] |
| Tank 2 | Mean SPL | $y = 36,452x - 4,193,465$ | 0.046 | 0.001 | 220 |
| Tank 2[a] | Mean SPL[a] | $y = 230,218x - 26,911,981$[a] | 0.048[a] | 0.470[a] | 11[a] |
| Tank 3 | Total no. of calls | $y = 689x + 35,106$ | 0.398 | 0.003 | 220 |
| Tank 3[a] | Total no. of calls[a] | $y = 2,052x - 506,339$[a] | 0.964[a] | 0.122[a] | 1[a] |
| Tank 3 | Mean SPL | $y = 31,294x - 3,533,082$ | 0.112 | <0.001 | 220 |
| Tank 3[a] | Mean SPL[a] | $y = 250,379x - 28,201,967$[a] | 0.997[a] | 0.036[a] | 1[a] |

**Notes.**
SPL, received sound pressure level in dB re 1 $\mu$Pa.
[a]Separate linear regression analysis was performed and did not include non-spawning events (i.e., when eggs collected were equal to 0).

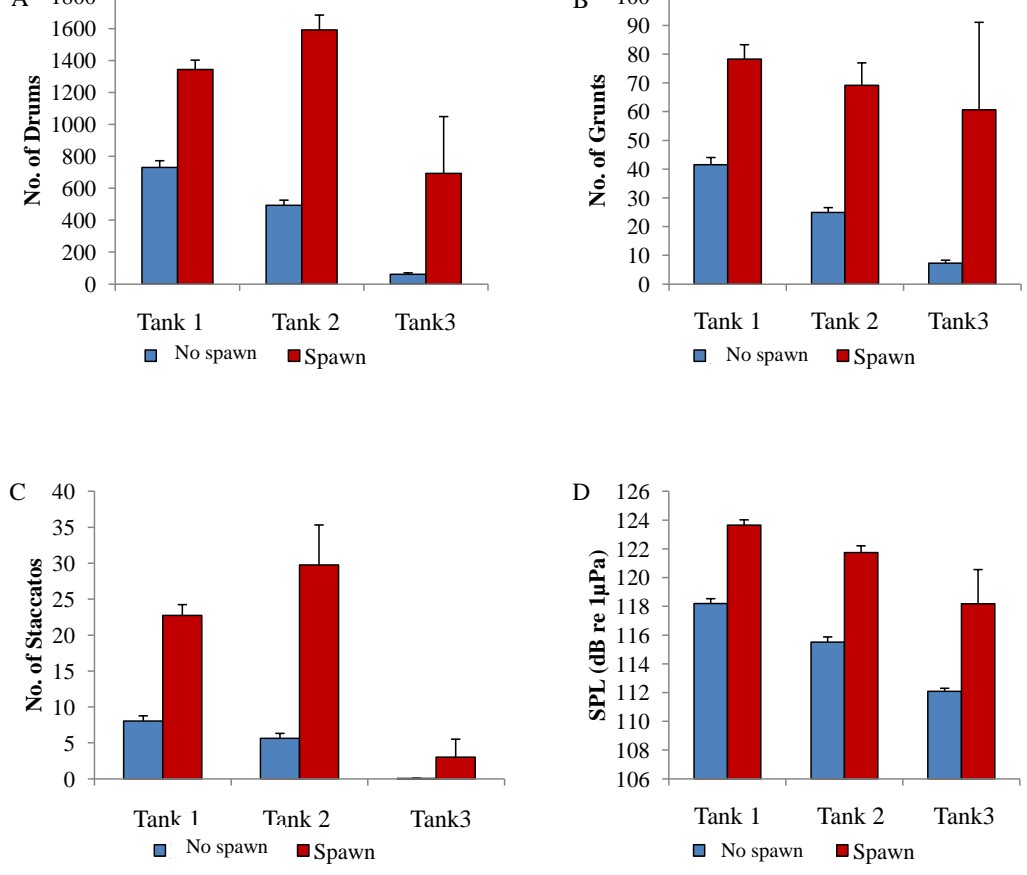

**Figure 8 Comparison between spawning & non-spawning.** A comparison of sound production of wild caught spotted seatrout (*Cynoscion nebulosus*) held in captivity for non-spawning and spawning periods. For each tank, (A) the number of drums; (B) the number of grunts; (C) the number of staccatos; and (D) the sound pressure level (SPL) were compared for non-spawning and spawning nights. No spawn, blue bar; spawn, red bar. Standard errors are reported.

locations (*Gilmore, 2003*). In our captive study, we grouped the "grunt followed by knocks" and "aggregated grunts" together because of their similarity in call structure and described these calls as "drums" to avoid confusion with the "grunt." We classified the "long grunt" as the "grunt", while the "staccato" strictly followed the description provided by *Mok & Gilmore (1983)*. In the present captive study, all of these call types (i.e., "drums," "grunts," and "staccatos") were associated with courtship behavior and spawning. We do realize a more thorough analysis of each call type would be beneficial, but this type of analysis is outside the scope of this paper.

Received SPL correlated with the sum of the drums, grunts, and staccatos (i.e., the total number of calls). Calculating SPL of the entire 'wav' file provides a way to calculate the overall acoustic energy. This metric is a function of the amount of calling, the number of pulses in a call, the call duration, the acoustic energy of each call, and how far away the fish is from the recorder. These findings are important, because in the wild, it is not feasible to count the number of calls in a chorusing aggregation due to overlapping calls. Calculating

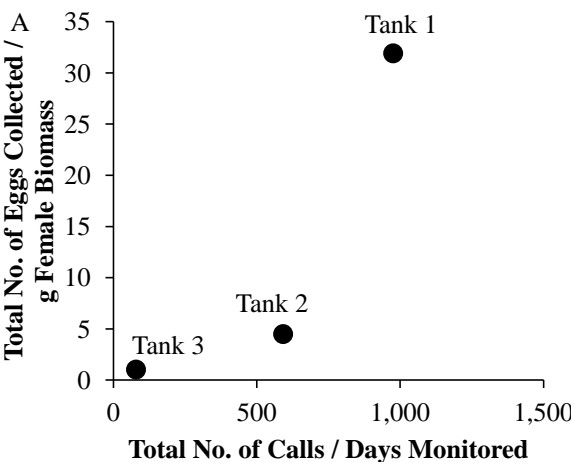

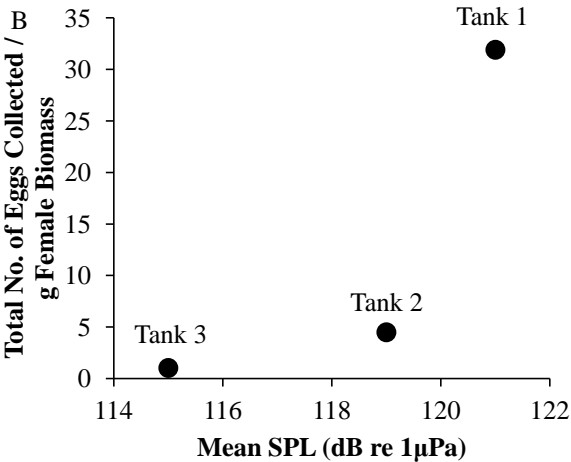

**Figure 9  Tank comparisons of sound production and spawning.** Tank comparisons of sound production and spawning for wild caught spotted seatrout (*Cynoscion nebulosus*) held in captivity. The total number of eggs collected per gram of female biomass versus (A) the total number of calls per days monitored and (B) the mean received sound pressure level (SPL) for Tank 1, Tank 2, and Tank 3. The total number of calls per days monitored was calculated by summing the total number of drums, grunts, and staccatos from 18:00 to 06:00 throughout the entire study period and then dividing this value by the number of days monitored. The mean SPL for each tank was determined by averaging all the 2 min SPLs from 18:00 to 06:00 over the entire study period.

SPLs becomes especially important for monitoring wild spawning aggregations over long time periods. The deployment of autonomous recorders generates thousands of files and having a MATLAB code that automatically calculates received SPLs is faster than manually reviewing acoustic files. One major drawback in determining received SPL in the wild is that the intensity depends on the distance from the spawning aggregation, which is often unknown. Another challenge is that the SPL measurement can contain acoustic energy

from other sources because the frequency range of a chorusing aggregation often overlaps with the frequency ranges of sounds produced by other marine organisms (i.e., snapping shrimp and other fish species) and boats.

## Seasonal and daily patterns in calling

We discovered that calling increased when the water temperature rose to around 27 °C and when the photoperiod increased to 14.5 h of light. These parameters mimicked the temperature and daylight hours observed during the summer, which is the spawning season in the southeast (*Luczkovich et al., 1999b*; *Roumillat & Brouwer, 2004*; *Luczkovich et al., 2008*). In North and South Carolina, sound production of spotted seatrout has been observed from May to September (*Riekerk, Tyree & Roumillat, 1997*; *Luczkovich et al., 2008*; *Montie, Vega & Powell, 2015*). Before the onset of spawning, circulating testosterone levels most likely increase and this change causes the sonic muscle to increase in weight. These hormonal and anatomical changes have been observed in weakfish as calling reaches its peak (*Connaughton & Taylor, 1994*; *Connaughton & Taylor, 1995*). An important point of this captive study is that it illustrates how deploying fixed, passive acoustic platforms (i.e., DSG-Oceans) can precisely define spawning seasons and relationships to water temperature. Tracking these sounds in the wild can provide exact spawning start and end dates, the length of the spawning season, and spawning frequency, which are necessary parameters in order to determine how climate change may shift or interrupt the timing of seasonal reproduction.

We found that wild caught spotted seatrout exhibited daily patterns of calling, which began when the lights turned off and reached maximum activity three hours later. Atlantic croaker, sand seatrout (*Cynoscion arenarius*), and red drum show similar calling patterns with sound production increasing at laboratory-simulated dusk (*Holt, Holt & Arnold, 1985*; *Montie, Vega & Powell, 2015*). In Charleston Harbor, South Carolina, spotted seatrout calling occurred from 18:00 to 22:00 h with a peak in the late evening (*Riekerk, Tyree & Roumillat, 1997*). In Pamlico Sound, North Carolina, spotted seatrout drumming activity began after sunset (21:00 h), peaked at 22:00 h, and ended at 23:00 h (*Luczkovich et al., 2008*). In Barataria, Caminada, and Eastern Timbalier Bay Systems of Louisiana, *Saucier & Baltz (1993)* showed that spotted seatrout sound production occurred from 17:00 to 01:00 h and that 92% of the drumming occurred between 19:00 and 23:00 h. The daily patterns of calling observed in the present study were comparable to the patterns reported in these wild studies. Sciaenids most likely spawn at dusk to limit the predation on eggs by many juvenile and adult fishes (*Holt, Holt & Arnold, 1985*).

In all tanks, rapid temperature changes within the simulated reproductive season affected calling (Fig. 4). Generally, rapid rises in temperature increased calling rates, while rapid declines in temperature decreased calling. Similarly, we found that wild caught red drum held in laboratory tanks exhibited similar temperature dependent behaviors (*Montie et al., 2016*). Other studies with different fish species revealed similar findings. In a study with tiger bass (*Terapon jarbua*), higher temperatures increased the amount of calling and the number of pulses in a call (*Schneider, 1967*). In oyster toadfish (*Opsanus tau*), increases in water temperature increased the number of courtship calls and their fundamental

frequency (*Fine, 1978*). In captive weakfish, rises in water temperature increased the mean SPL, the mean number of pulses, and the mean frequency of calls (*Connaughton, Taylor & Fine, 2000*). Our findings and past studies indicate that changes in water temperature as low as 2 °C can affect calling rates, which could in turn affect the ability of a reproductively active female to find a spawning aggregation. Thus, climate change could have important implications on the acoustic communication of fish, spawning, and reproductive output.

In the wild, there is some evidence that spotted seatrout calling and spawning are associated with the lunar cycle. *Gilmore jr (1994)* discovered that spotted seatrout calling occurred most often on the full moon or within three to four days after the full moon. However, our preliminary studies have indicated different patterns than the findings observed by *Gilmore jr (1994)*. Since 2013, our lab has been monitoring the underwater soundscape at three stations in the May River, South Carolina using DSG-Ocean recorders. In the May River, our studies have indicated that longer chorusing episodes with earlier start times occur on the second and fourth quarter lunar cycles rather than on the third cycle (i.e., full moon) as observed by *Gilmore jr (1994)* (E Montie, 2016, unpublished data). In the present captive study, a lunar light cycle was not included in the simulated reproductive season. Not including this natural rhythm may have altered the natural spawning patterns observed in the wild. This may partly explain why sound production and egg deposition were not more tightly correlated in the present captive study.

## Sound production influences spawning success

Spotted seatrout were more likely to spawn when male fish called more frequently. In addition, there was some evidence that spawning was more productive when rms SPLs were higher. In Tanks 1 and 3, we discovered a statistically significant positive relationship between SPL and egg deposition. In our captive red drum study, we found a much stronger relationship between the amount of calling and the number of eggs collected (*Montie et al., 2016*). In the present study, our findings did not indicate that the call type or call composition played a differential role in spawning success. However, we found that the call structure was important in spawning success of wild caught red drum (*Montie et al., 2016*). In that study, we demonstrated that the number of pulses in a call was higher and the mean call duration was longer on evenings when spawning did occur (*Montie et al., 2016*). In the present study, we did not examine, in detail, how changes in call structure affected spawning success. It would be interesting to examine the acoustic characteristics of each call type and determine if these characteristics are different between non-spawning and spawning events. These characteristics could include frequency range, peak frequency, sound pressure level, call duration, number of pulses in a call, pulse duration, and pulse interval for each of the call types. However, this type of analysis is outside the scope of this paper and will be the focus of future work. A more thorough examination may find that the complex repertoire of calls of spotted seatrout may affect spawning success.

We observed that sound production varied among tanks. More calling was detected in Tanks 1 and 2 as compared to Tank 3 (Table 2; Fig. 4). Tanks 1 and 2 contained more males than Tank 3, while Tank 3 contained more females (Table 1). Only male spotted seatrout have a sonic muscle and produce sound, which explains why more calling was

detected in Tanks 1 and 2 as compared to Tank 3. In addition, spotted seatrout in Tanks 1 and 2 spawned more often and produced more eggs per gram of female biomass than seatrout in Tank 3, despite having close to twice the number of females in Tank 3 (Table 1; Fig. 9). These findings may indicate that having more males that are acoustically active in a spawning aggregation are key factors in enhancing reproductive output and sustaining populations.

Other studies performed in captive environments have qualitatively demonstrated a relationship between calling and reproductive events in red drum, weakfish, and white seabass (*Atractoscion nobilis*) (*Guest & Lasswell, 1978*; *Connaughton & Taylor, 1996*; *Lowerre-Barbieri et al., 2008*; *Aalbers & Drawbridge, 2008*). In wild studies, plankton tows and acoustic recordings performed on the same night and location have revealed an association between sound production and spawning (*Mok & Gilmore, 1983*; *Saucier & Baltz, 1993*; *Connaughton & Taylor, 1995*; *Luczkovich et al., 1999a*). In black drum, silver perch, and spotted seatrout in Indian River Lagoon, Florida, peak calling occurred between 17:00 and 22:00 h, which coincided with the presence of eggs and larvae in the water column (*Mok & Gilmore, 1983*). In studies with wild spotted seatrout in the Barataria, Caminada, and eastern Timbalier Bay systems of Louisiana, tows downstream of drumming aggregations contained two to three times more eggs in comparison to tows upstream (*Saucier & Baltz, 1993*). In studies with weakfish and silver perch aggregations in Pamlico Sound, North Carolina, SPLs of aggregations positively correlated with the presence of "sciaenid-type" eggs (*Luczkovich et al., 1999a*).

What exact role does sound production play in the spawning process of sciaenids? Male calling most likely aids the attraction of a gravid female to a spawning location as suggested by *Connaughton & Taylor (1996)*. Larger aggregations of males and more calling increases the overall SPL of the chorus, which can propagate outward over larger distances from the source and attract females from further locations. In addition, drumming may contain information regarding male fitness (*Connaughton & Taylor, 1996*). In our study with captive red drum, spawns were more productive on evenings when calls were longer in duration and contained more pulses (*Montie et al., 2016*); however, this association was not found in the present study with captive spotted seatrout. In some sciaenids, the variability in call structure may indicate that males compete with each other to be chosen by females (i.e., intersexual selection). It is possible that the good gene model in intersexual selection (i.e., females choose males based on physical characteristics or abilities that may display some genetic advantage) plays a major role in courtship behaviors and spawning success. For example, it has been shown that female gray tree frogs (*Hyla versicolor*) prefer to mate with male frogs that advertise long mating calls (*Welch, Semlitsch & Gerhardt, 1998*). Because offspring fathered by long-calling males outperformed their half-siblings fathered by short-calling males (i.e., with regards to larval survival, growth, and time to metamorphosis), *Welch, Semlitsch & Gerhardt (1998)* concluded that the duration of a male's mating call is indicative of the male's overall genetic quality. More research is necessary to better understand the formation, maintenance, and dynamics of spotted seatrout spawning aggregations and the exact function of male calling.

## CONCLUSIONS

Spotted seatrout were more likely to spawn when male fish called more frequently. Yet, there were several times during the recording period when there was calling but no spawning occurred. Some aspects of spotted seatrout reproductive biology may explain this observation. Most estimates of spawning frequency suggest that female spotted seatrout spawn once every 4–5 days, as reviewed by *Brown-Peterson (2003)*. In addition, there is some evidence that spotted seatrout calling and spawning are associated with the lunar cycle, which was not controlled for in the present study.

Nonetheless, our findings indicate that we can use acoustic metrics, with confidence, to predict spawning potential. Passive acoustics can be used to find the location of spawning aggregations, determine spawning start and end dates, and possibly estimate maximal values of spawning frequency within a reproductive season. These findings are significant because plankton tows may not accurately reflect spawning locations since egg capture is likely affected by predator activity, water currents, and tow efficiency. Instead, passive acoustics could be used to monitor spotted seatrout reproduction. Future studies can use this captive study as a model to record the estuarine soundscape precisely over long time periods to better understand how human-made stressors (e.g., climate change, noise pollution, and chemical pollutants) may affect spawning patterns.

## ACKNOWLEDGEMENTS

We thank the staff of SCDNR for husbandry care of spotted seatrout. We would also like to thank the following students and staff from USCB for their help in collection of data, analysis, and editing: Matt Hoover, Rebecca Rawson, Steven Vega, Michael Powell, Alishia Zyer, and Dr. Brian Canada.

### Funding

This work was supported by an ASPIRE-1 grant from the University of South Carolina (USC). Additional funding came from the National Institute of General Medical Sciences (NIGMS) grant no. P20GM103499 of the National Institute of Health (NIH), the University of New Hampshire and the US Department of Commerce/NOAA grant number NA09NOS1490153, the USCB Sea Islands Institute, Beaufort County Storm Water Utility and the Palmetto Bluff Conservancy. Spotted seatrout production support was provided by South Carolina Saltwater Recreational License Funds and the State of South Carolina. The funders had no role in study design, data collection and analysis, decision to publish, or preparation of the manuscript.

### Grant Disclosures

The following grant information was disclosed by the authors:
University of South Carolina (USC).
National Institute of General Medical Sciences (NIGMS): P20GM103499.

University of New Hampshire and the US Department of Commerce/NOAA: NA09NOS1490153.

USCB Sea Islands Institute, Beaufort County Storm Water Utility.

Palmetto Bluff Conservancy.

South Carolina Saltwater Recreational License Funds and the State of South Carolina.

## Competing Interests

The authors declare there are no competing interests.

## Author Contributions

- Eric W. Montie conceived and designed the experiments, performed the experiments, analyzed the data, contributed reagents/materials/analysis tools, wrote the paper, prepared figures and/or tables, reviewed drafts of the paper.
- Matt Hoover and Christopher Kehrer analyzed the data, prepared figures and/or tables, reviewed drafts of the paper.
- Justin Yost and Karl Brenkert performed the experiments, analyzed the data, reviewed drafts of the paper.
- Tim O'Donnell analyzed the data, wrote the paper, reviewed drafts of the paper.
- Michael R. Denson conceived and designed the experiments, performed the experiments, contributed reagents/materials/analysis tools, wrote the paper, reviewed drafts of the paper.

## Animal Ethics

The following information was supplied relating to ethical approvals (i.e., approving body and any reference numbers):

This work was part of the SCDNR restocking program for spotted seatrout and did not require an IACUC. No fish were harmed or sacrificed in this study.

## Data Availability

The raw data has been supplied as a Supplementary File.

## Supplemental Information

Supplemental information for this article can be found online at http://dx.doi.org/10.7717/peerj.2944#supplemental-information.

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
