# Peer review of "Acoustic monitoring indicates a correlation between calling and spawning in captive spotted seatrout (Cynoscion nebulosus)"

_PeerJ, doi:10.7717/peerj.2944_

## Round 0.1 · original submission · Major Revisions

Please revise according to the reviewers' comments. Please pay particular attention to the serious concerns raised by Reviewer 2. You will need to show that these concerns are not, in fact, as serious as they appear in order for the paper to be published. Please be aware that the revised paper may be sent back to that reviewer to ascertain that you have adequately addressed the problems the reviewer identified.

·

Basic reporting

fine

Experimental design

fine

Validity of the findings

fine

Additional comments

Montie et al provide a detailed study relating sound production to number of eggs produced in a captive situation over a season that appears to mimic what happens in the field. This allows the authors much greater control since they know the number of fish and can quantify the number of eggs produced without losses due to predation or dilution that would occur in the field.
The study at first appears to be well written, but the tremendous redundancy started to wear on me. I believe the authors will increase the number of readers who stick with the paper to the end by shortening the paper. I will give a number of examples later, but for now if something applies to multiple sounds or tanks, say it once at the beginning of a paragraph rather than repeating the same phrases multiple times. We don’t need to be told every time it is mentioned that tank 3 had fewer males.
Commas should be inside the quotation marks.
Although this paper focuses primarily on numbers of sounds and correlations with temperature, photoperiod and number of eggs laid, the authors should consider the communication system in a little more detail. The variety of sounds produced by the spotted sea trout far exceeds that produced by congeneric weakfish. I realize the authors are limited by not being able to witness interactions between fish, but some thought should be devoted to this complexity since they have extensive data. My thought is that different sounds are likely produced in the same situation, and perhaps the variety is attractive to females. I realize this suggestion goes beyond the data, but more should be said in the discussion about the complexity, and this would be a great avenue to focus on in future work.
Sounds could be described more fully. We read that the staccato call has a very short inter-pulse interval, and the drum call has a short inter-pulse interval. I would not expect the authors to measure the pulse duration and interval on every sound in the study, but these should be quantified on at least a sample. The acoustic energy ranging from 50 to 1000 Hz is mentioned three times in the paragraph starting on line 254. Say it once at the beginning and then focus on differences in the sound types, which would be a reflection of input from the central nervous system. Perhaps issues of more complete description and implications of call complexity can be avoided if the authors says they will be considered elsewhere. At the least get rid of short and very short. I am thinking the authors should either do more or less on the communication aspect, and there is enough information in the data presented to justify less. Just don’t be in the middle.
Figure 2A shows a sequence of calls produced in a 10 sec period, which is nice to illustrate the complexity of different sounds recorded in a short time span. I would however prefer to see a better example of a staccato call (whether from a continuous time interval or not) since the first selection includes sounds from two and possibly three fish. The example of the staccato is from the sound in the background from a fish further from the hydrophone than the more obvious pulses. The multiple callers should also be mentioned since a casual reader might think that the staccato came from a single individuals. I am wondering if the closer fish might actually be producing a drum call?
I don’t hold it against the authors, but the slow decay shown in Fig 2D likely indicates tank effects on the sound, and the authors should definitely remove their statements about no tank artifacts.
L. 47. Replace significant with important.
L. 168 Remove exact; the same dimensions says it. L. 171 Remove that is all. L. 175 Recorded for two minutes every 20 minutes is sufficient without further examples unless it is important for us to know the sounds were recorded from 12:00 to 12:02 rather than from 12:02 to 12:04. L. 176 Remove a total of. L. 185. Mok and Gimore’s classification has already been given and is mentioned again in the discussion. Not necessary to repeat here. L. 206 factor not fator. L. 231 Replace the most significance with highest. Further in this paragraph delete repeated statements of multiple categorical dependent variables as 0 or 1. Say it once in the paragraph. L. 251 Remove nomenclature published in other. L. 269. Replace least amount of calling with fewest calls. If you have to mention tank 3 having fewer males, once is enough per section. L. 277 More prevalent than what? L.288 Replace spawning with courtship. L. 292 Replace most significant with greatest. L 294 Replace did reproduce with reproduced and remove a bunch of dids elsewhere in the ms. L. 302.Delete we discovered that. L. 304 Here and elsewhere delete calculated over the entire monitoring period. L. 305 Delete sentence starting However,…L. 312 Delete that we observed. Eric seems to fear that if he does not specify everything precisely in each sentence, the reader won’t get it. In reality this over specification interferes with appreciation of the ms.
L. 329. Replace sentence with: Received SPL correlated with the sum of the drums, grunts and staccatos (i.e the total number of calls).
L. 334 Replace sound with fish. L. 335 Sentence starting On the other hand is tedious and states that sound pressure level expresses amplitude. Come on! L. 338 Replace: if the plan is to monitor with for monitoring. L. 341. Delete “much” and “having an observer, replace and with to. L. 351 Replace shifted with increased. L. 352 Replace which is the time period spotted seatrout spawn in the Southeast with which is the spawning season in the southeast (note small s). L. 369 occurred rather than was shown to occur. L. 370 Replace with peak sound production happening with a peak. L. 378 Holt suggested rather than It is suggested by Holt. L. 380 Delete studies that have performed. L. 385. Replace In a different but similar study with Similarly. L. 411 Delete In this study, we found that. L 414 Again say Similarly.
L. 421. In frogs there are species with burst and protracted spawning periods. There is no indication that sound becomes less important in species with shorter seasons. I think you would have a tough time defending your hypothesis. Again consider congeneric weakfish that have a much simpler vocabulary even though they have a long spawning season also.
L 447. Do you really want to say that tows revealed tows?
L. 449 and 452. Delete it was found that. L 478 Delete We found. L. 481. Should you put in female before the trout?
This paper provides excellent and extensive data on long term calling patterns and egg production of an important fish species that will be relevant to the burgeoning field of passive acoustics as well as acoustic communication. The writing should be simplified and shortened as a favor to the reader. Some of my suggestions on adding information on communication may not be totally necessary considering the scope of the paper. I do note that Eric is an excellent position to examine the complex calls of speckled trout if it is too much for this paper.
Michael

Reviewer 2 ·

Basic reporting

The manuscript was well written in clear and unambiguous English language. The introduction and background were thoroughly researched, although the authors only cited references related to the reproduction associated sounds of Sciaenid species. It would be nice to also include some comments on similar studies involving species from other families (e.g; haddock, cod, rockfish etc) to further highlight the significance of the current paper.

Figures in general are of good quality and clearly highlight the intended points. My only comments are;
i) Figure 2 A (spectrogram), E, F & G (power spectra), which should include (in the figure caption) the information on the FFT size used to plot them. This is critical for the reader to understand the frequency resolution of spectrograms and power spectra.
ii) Add one more curve (power spectra) of the control SPL (i.e. background noise in tank without the fish) in Figure 2E, F & G. This would clearly illustrate the relative increase in SPL due to fish callings.
iii) An additional figure showing a schematic diagram of the placement of the acoustic logger relative to the tank dimensions and fish position would be good to clarify the recording setup.

Experimental design

A) The experimental design was aimed to demonstrate a clear association between the calling patterns and spawning behavior in spotted seatrout. The idea was that such calling pattern could be used as a proxy to identify spawning events for the particular fish species. However, the experiment was a serendipitous one (i.e; concurrent with other SCDNR studies) which in a way was not ideal (as readily admitted by the authors). Furthermore there is a risk that the experiment may not be bioacoustically sound. In conjunction with the fact that recording underwater sounds of aquatic animals in tank confinements more so within a building is technically difficult and challenging, I find the following major concerns glaring in the current paper;

1. The effect of the recording tank’s resonance (resonant frequencies) was not mentioned anywhere in the paper therefore I assumed was not considered.
- In most cases, tank resonant frequencies may render recordings of underwater signals in tanks useless when their frequencies overlap or close to each other. In such instances, acoustic measurements such as the dominant frequency and the sound pressure level (SPL) of the signal are contaminated and therefore are not correctly measured.
- I strongly suggest the authors refer to Akamatsu et al. (2002) for further clarification on this problem and follow its suggestion on ways to solve this critical issue.
- Following this, the authors must clarify in the paper that tank resonance was not an issue and that it did not affect the fish sound recordings.

2. Since there was no mention in the paper about the elimination efforts of noise from extraneous sound source (water pumps, water flow, aeration, building vibration, ac current and other electrical as well as mechanical equipments) other than the fish’s during recordings, I presumed that these noise were simultaneously recorded therefore potentially contaminating the fish sound signal.
- Noise from extraneous sound sources in laboratory settings as mentioned in the above, typically consisted of high amplitude low frequency sounds below 100 Hz.
- Therefore for indoor tank recordings, it is common practice to filter out this low frequency noise during sound recording or not to include sounds in this frequency range during the signal processing analysis or measurement of sound power.
- I strongly suggest that the authors consider this issue and measure sound power within a narrower frequency band that covers the known bandwidth of the sound produced by this species (e.g; >100 Hz or 200-1000 Hz)
- Unfortunately I consider measuring the loudness of fish calls together with its loud background noise flawed even more so if it includes measuring sound power in frequencies outside the bandwidth of the fish sound.

3. I could not find the write-up about recording a control sound recording of the tank setup (background noise without the fish in the tank). This is a good practice to enable the measurement of elevated SPL due to fish calls by simple subtraction.

B) With regards to recording schedule, I personally feel that periodic recording of 2 mins/20 mins or a total of 6 mins/hr is too short of a recording to accurately identify temporal patterns of fish sounds that is transient in nature. Therefore a slightly longer recording duration or even better, continuous recording would be best. However I understand that the experiment did entail a rather long-term schedule, which would otherwise exhaust the storage of the recorder should the recordings be longer. Perhaps for future work longer recording durations can be achieved by reducing the sampling rate of the loggers.

Validity of the findings

a) In the “Description of Spotted seatrout calls in captivity” section (results and discussion), the authors put great emphasis to report that all fish calls in the current study were comparable to those of past studies on the same species by various researchers (cited). I am not sure how the comparison was made since the detailed acoustic characterics (eg; sound duration, pulse number/duration/period/interval, dominant frequency, frequency bandwidth etc) of the current study’s fish calls was not presented in the paper. I would find it rather premature to assume similarity of their fish sound with past studies fish sound merely from general description of the sound without strictly/statistically comparing their detailed acoustic characteristics. I feel it is necessary to conduct a detailed acoustic characterization (with sufficient sample size of each sound type) of the fish calls in the current study and include in the paper a report/table that summarizes the respective acoustic characteristics.
b) Regarding SPL measurements in the study, I like to revert to my previous comments on the experimental design. Unfortunately without addressing and acting on the issues (tank resonance, noise contamination and analysis bandwidth) that I have mentioned earlier, I feel that all SPL measurements meant to represent those of fish calls were flawed and therefore not appropriate for inclusion in the paper. This flaw implicated several analysis results (linear regression analysis and correlation analysis) and results in Table 2,3 & 4 as well as Fig 2 (D,E,F), 7 & 8. Sorry….

c) In the “Seasonal and daily patterns in calling” section, the findings are clearly evident from the results of the experiments and the authors aptly discussed the likely explanations behind it. The temporal patterns observed in this study partially concur with temporal calling patterns in the wild for the same species as reported in other studies with some differences clearly explained by the authors.

d) The only comment for this section is on the statement (line 363-364) regarding how ability to track fish calls like the findings from this study could play a role in understanding effect of climate change on spawning time of fish which I feel is an over-speculation. So is the statement in line 395-396.

e) In the “Sound Production Influences Spawning Success” section, again statements linking the “flawed” SPL measurements to spawning productivity (line 412-414) is invalid and need to be excluded until the validity of the SPL measurement is verified.

f) In line 428-436, the authors compared calling and spawning activity among tanks and acknowledge that the different measurements are likely due to differences in fish composition. I feel that such inference could not be derived from this study due to the lack of treatment (fish composition) replication. In relation to this, the statement in line 434-436 is also unfounded and need to be excluded from the text.

g) In the paragraph beginning from line 456 onwards, the authors attempted to explain the exact role of sound production on spawning which was not supported by any findings nor is it an objective of the current study. I feel that such discussion is irrelevant thus need not be included in the paper.

Additional comments

- Bioacoustical experiments are ideally conducted independently where experimental designs can be thoroughly thought out and specific measures and control especially those regarding reducing extraneous noise can be put in place. Serendipitous type bioacoustics experiment such as the one in this reviewed paper is risky and is subjected problem that could be detrimental to the validity of the bioacoustical data. Nonetheless understanding and acknowledging the limitation of such experimental design and taking appropriate steps to reduce the severity of the negative factors could help make the experiment acceptable. Some of these steps were suggested in my review.
- Even if the SPL measurement is deemed to be flawed and not used, I still think that the calling activity data is suffice to support the study hypothesis and show that the increased in male fish sound production and spawning events are correlated.

---

## Round 0.2 · accepted · Accept

Thank you for your revisions and especially for clarifying the issues raised by the more critical reviewer.

·

Basic reporting

Nice study with abundant data

Experimental design

Fine considering the authors were dealing with large fish in captivity

Validity of the findings

Valid

Additional comments

I am happy with the revisions;I think the authors did a good job of answering criticisms, particularly of the second reviewer.

Note that typically periods and commas are placed inside quotation marks.
L 184. Remove "do."
L. 275 Two periods. Remove one of them.

Reviewer 2 ·

Basic reporting

The manuscript was well written in clear and unambiguous English language. Additional relevant information was added that help emphasized the importance of this study.

Experimental design

The experimental design was aimed to demonstrate a clear association between the calling patterns and spawning behavior in spotted seatrout. The idea was that such calling pattern could be used as a proxy to identify spawning events for the particular fish species. The experimental design appear sound with detailed clarification by the authors eliminating the potential issues inherent to tank recordings.

Validity of the findings

Findings of the study are valid based on good data collection and analysis.

Additional comments

The main concern in my previous review of the manuscript was on:

1) The experimental design of the study namely on the effects of tank resonance and extraneous noise on the sound recordings.

- The comment was not about the experimental design being faulty but rather on insufficient clarifications by the authors that inherent problems of recording sound in tanks was appropriately addressed therefore was not an issue in their study. In the revised version of the manuscript, the authors have clearly shown from model calculations and observations that tank resonance had no major effects on the sound recordings (line 184-208). As for extraneous noise, the authors have clarified that these noise were minimal (due to the sophisticate SCDNR tank facilities) and well below the sound levels of seatrout calls (line 243-245). This clear explanation together with relevant proofs that both the effects mentioned above had been appropriately addressed is adequate to validate the related SPL measurements.

2) The description of seatrout calls.

- The “description of seatrout calls” section in the manuscript was merely to relate that the captive fish calls in the study conformed to the type of calls produced by the species as reported in past studies. Detailed acoustic characterization of the calls was outside the scope of the study and was clearly clarified by the authors (line 296- 307).

3) Minor amendments to figures and captions.

- The authors have made amendments by adding figure 1B and C to clarify the placement of the hydrophone and relative distance of fish to hydrophone. This is useful for the reader to visualize the underwater sound recording. The caption for Figure 2 and 3 was also amended to include FFT size and frequency resolution information. Figure 3 was amended to include plots of background noise together with the fish call plots that clearly helps with visualization of the elevated SPL due to fish calls.